# Tetanus in Romania—Trends and Challenges

**DOI:** 10.3390/microorganisms13071654

**Published:** 2025-07-12

**Authors:** Andreea Marilena Păuna, Ștefan Eduard Mîinea, Bianca Georgiana Enciu, Daniela Pițigoi, Anca Mirela Sîrbu, Rodica Popescu, Carmen Daniela Chivu, Carmen-Cristina Vasile, Maria Dorina Crăciun

**Affiliations:** 1Department of Epidemiology I, Carol Davila University of Medicine and Pharmacy, 021105 Bucharest, Romania; andreea.pauna@umfcd.ro (A.M.P.); stefan-eduard.miinea@drd.umfcd.ro (Ș.E.M.); carmen-daniela.chivu@umfcd.ro (C.D.C.); carmen-cristina.vasile@rez.umfcd.ro (C.-C.V.); maria.craciun@umfcd.ro (M.D.C.); 2Medico-Military Institute, 010919 Bucharest, Romania; 3Department of Epidemiology II, Carol Davila University of Medicine and Pharmacy, 050463 Bucharest, Romania; bianca.enciu@umfcd.ro (B.G.E.); anca-mirela.sirbu@umfcd.ro (A.M.S.); 4National Centre for Communicable Diseases Surveillance and Control (CNSCBT), National Institute of Public Health, 050463 Bucharest, Romania; rodica.popescu@insp.gov.ro; 5National Institute for Infectious Diseases “Prof. Dr. Matei Balș”, 021105 Bucharest, Romania; 6Clinical Emergency Hospital for Children “Grigore Alexandrescu”, 011743 Bucharest, Romania

**Keywords:** tetanus, surveillance, fatality, vaccination, tetanus toxoid booster doses

## Abstract

Tetanus is a life-threatening, vaccine-preventable disease caused by tetanospasmin and tetanolysin, which are potent neurotoxins produced by *Clostridium tetani*, an anaerobic, spore-forming bacterium. Due to the widespread presence of spores in the environment, the disease cannot be eradicated. However, global tetanus prevention initiatives have contributed to a significant decline in tetanus incidence worldwide. Aiming to present the tetanus trends in Romania, we conducted a retrospective analysis of the tetanus surveillance data. During the study period (2010–2023), 97 cases of tetanus were reported in Romania (median: 6.5; IQR: 5–7) with an average incidence rate of 0.03 per 100,000 inhabitants (95% CI: 0.02–0.04; range: 0.01–0.09). The highest incidence rates were recorded among people aged 1 to 14 years old (0.09 per 100,000 inhabitants, 95% CI: 0.06–0.13; range: 0.00–0.20), male (0.05 per 100,000 inhabitants; 95% CI: 0.03–0.06; range: 0.03–0.12), and from rural areas (0.05 per 100,000 inhabitants; 95% CI: 0.03–0.08; range: 0.01–0.17). A decline in the number of tetanus cases of 7% by year was observed, which is supported by the statistical analysis showing a *p*-value of 0.005 (IRR: 0.93; 95% CI: 0.88–0.98). However, the same decline in tetanus incidence was not supported by the statistical analysis (IRR: 0.93; 95% CI: 0.44–1.98; *p* = 0.9). Forty-seven tetanus deaths were recorded, with an average case fatality ratio of 42% (95% CI: 25.62–57.92; range: 0–100), showing a decreasing trend of 9% by year (IRR: 0.91; 95% CI: 0.89–0.93). Although the annual number of tetanus cases in Romania has shown a slight downwards trend, its situation has remained relatively stable, as shown by the tetanus incidence. Moreover, the case fatality rate continues to be high. Therefore, our study emphasizes the importance of achieving high vaccination uptake among children and adolescents, raising awareness of the importance of booster doses in adults, and improving the management of tetanus-prone wounds.

## 1. Introduction

Although it is a vaccine-preventable disease that has been described since ancient times, without human-to-human transmission, seemingly paradoxically, tetanus continues to remain a public health problem. This phenomenon is due to the particularities of the epidemiological process and the characteristics of the etiological agent.

The first documented description of the disease dates back to the ancient Egyptian period, ca. 1500 BC, with manifestations such as rissus sardonicus and trismus being presented in the Edwin Smith Papyrus surgical section. Later, in his work *Epidemics*, Hippocrates (460 BC–370 BC) provided a more detailed description of tetanus [1].

The etiological agent, *Clostridium tetani* (*C. tetani*), is a Gram-positive, obligate anaerobic bacillus. Tetanus is caused by plasmid-containing neurotoxin-producing strains. The bacillus can persist for long periods in the environment as resistant spores. Although ubiquitous, spores are more commonly found in organic-rich soil in warm, humid climates [2,3].

The bacillus is excreted into the environment through the faeces of various animal species, including humans. The most affected animal species are horses, guinea pigs, monkeys, sheep, mice, and goats, while cats and dogs are less susceptible, and birds are resistant [4,5].

A potentially tetanus-prone wound is a lesion contaminated with spores that provides an anaerobic environment. Therefore, the portal of entry is a wound that is often minor and may go unnoticed, such as a deep puncture by sharp wooden or metal objects, thorns, lacerations, open fractures, surgical interventions performed under improper conditions, dental infections, inoculation with a contaminated syringe, animal bites, or insect stings. For neonatal tetanus, the portal of entry is the umbilical cord wound produced by the use of non-sterile instruments and dressings [3,6].

At the inoculation site, under anaerobic conditions, tetanus spores germinate and release the toxins tetanospasmin and tetanolysin. The toxins inhibit neurotransmitter release in the brainstem and spinal cord [2,3].

After an incubation period of 3 to 21 days or more, a clinical picture characterized by rigidity, muscle spasms, lockjaw, rissus sardonicus, painful generalized muscle spasms and arched back spasm (opisthotonus), and respiratory distress can be observed [3,7].

Four clinical forms of tetanus are described: generalized tetanus, the classic and most severe form of the disease, representing approximately 95% of cases; localized tetanus, a rare form, with muscle rigidity limited to the wound area; cephalic tetanus, resulting from a wound in the facial region; and neonatal tetanus, a form of generalized tetanus that occurs in newborns of unimmunized mothers when birth occurs under inappropriate conditions [3,7].

In severe, complicated forms, muscle spasms can cause fractures and tendon avulsions. Additionally, kidney damage, hemodynamic disturbances, and arrhythmias may occur.

Complications related to long-term hospitalization in an intensive care unit such as ventilator-associated events including ventilator-associated pneumonia, pulmonary thromboembolism, and catheter-associated urinary tract infection may occur. The therapeutic management of these complications represents a challenge due to the increasing phenomenon of antibiotic resistance [7,8].

The case–fatality ratio ranges from 10 to 90%; it is highest in infants and the elderly. It varies inversely with the length of the incubation period and the availability of intensive care [6,9].

Most cases of tetanus occur in incompletely vaccinated people or people with an unknown vaccination history who present a high-risk lesion [10].

Groups of people considered at high risk for tetanus include workers who come into contact with soil, people who work with sewage and drinking water supply systems, farm workers, military personnel, police officers, adults with diabetes, elderly individuals, people experiencing homelessness, persons who inject drugs (PWIDs), and travellers to areas endemic for tetanus [6].

To ensure continued protection against tetanus and diphtheria, booster doses (dT or dTap) should be administered every 10 years throughout life [11].

Women of childbearing age are recommended to be revaccinated every 10 years with dT, dTap, or ATV (adsorbed tetanus vaccine) and in the last trimester of pregnancy for the prophylaxis of maternal and neonatal tetanus [6].

Before 1924, when the tetanus toxoid was developed, tetanus had high morbidity and mortality worldwide [12].

The use of tetanus toxoid-containing vaccines, which are included in the Expanded Programme of Immunization of the World Health Organization (WHO), has contributed to a decrease in its incidence in the post-vaccination era from 1.54 to 0.39 per 100,000 [13].

Additionally, the maternal and neonatal elimination initiative launched by the World Health Assembly in 1988 contributed to the achievement of neonatal tetanus elimination status in 45 of 59 countries (as of March 2018), which is defined as fewer than 1 case per 1000 livebirths in every district of the country. According to the WHO estimates, 24,000 newborns died in 2021 because of neonatal tetanus, an 88% decrease from 200,000 in 2000 [14].

Despite these initiatives, tetanus still imposes pressure in low- and middle-income countries in Asia and sub-Saharan Africa, where its burden is often underestimated because of the limitations of the existing surveillance systems. In contrast, in high-income countries with high vaccination coverage, tetanus is a rare disease, occurring especially among some risk groups such as elderly individuals, PWIDs, and diabetes patients [15]. In Europe, tetanus is a rare disease. According to the surveillance data collected from 2018 to 2022 by the European Centre for Disease Prevention and Control (ECDC) within The European Surveillance System (TESSy), a total of 289 tetanus cases were reported. Italy, Poland, and Romania reported the highest number of cases, while Slovenia recorded the highest incidence rate. People over 65 years of age and women were the most affected. However, comparisons between countries are limited considering the differences among the surveillance systems. Additionally, underreporting or under-ascertainment of tetanus cases is possible, particularly since partially immunized patients may present with very mild disease [16].

In Romania, the diphtheria–tetanus–pertussis (DTP) vaccine was introduced in 1961. Initially, the vaccine was offered to children and adolescents as part of the National Vaccination Program.

Subsequently, the National Vaccination Program provided a tetanus-containing vaccine for pregnant women.

Since December 2008, the vaccine with the acellular pertussis component (DTaP) has been used. According to the national vaccination schedule, at present, primary vaccination against tetanus consists of three doses of combined diphtheria–tetanus–acellular pertussis–hepatitis B–inactivated poliovirus–Haemophilus influenzae type b vaccine (DTaP-HBV-IPV-Hib) at 2, 4, and 11 months, followed by two booster doses, one of DTaP-IPV at the age of 5–6 years and the second of dTap at 14 years [17,18].

From 1 December 2023, the dTap vaccine is fully reimbursed for pregnant women (vaccination that was previously provided within the National Vaccination Program) and for all adults, a booster every 10 years [19,20]. However, vaccination coverage is suboptimal [21]. The protocol for the management of potentially tetanus-prone wounds elaborated by the National Centre for Communicable Diseases Surveillance and Control (CNSCBT) within the National Institute of Public Health (NIPH), the institutional structure coordinating tetanus surveillance in Romania [6], indicates vaccination in selected cases.

We aim to describe the evolution of tetanus in Romania over a period of 14 years, considering the placement of Romania in the top positions among the European Union/European Economic Area (EU/EEA) countries due to the high morbidity and mortality from tetanus.

## 2. Materials and Methods

### 2.1. Study Design, Data Sources, and Statistical Analyses

We conducted a retrospective study using tetanus national surveillance data from 2010 to 2023. We collected demographic, clinical, and epidemiological data and the tetanus vaccination status of the cases. We performed a descriptive and comparative analysis of the main demographic and epidemiological indicators used for describing tetanus evolution: the number of cases and deaths, the incidence rates (general and specific—by age group, sex, place of residence, county) with 95% confidence intervals (95% CIs), the case fatality ratio, and the mortality rate. Incidence rate ratio (IRR) was calculated to highlight trends in tetanus situation over time. Four age groups were described: 1–14 years, 15–24 years, 25–64 years, and 65 years and over.

We used as a denominator the population by domicile on 1 July, which is publicly available on the National Institute for Statistics website [22]. Additionally, we used linear regression models to assess the differences among different groups as well as the impact of vaccination coverage at the age of 12 months on tetanus incidence, using the vaccination data publicly available on the NIPH website and the CNSCBT’s old website, respectively [21,23]. Moreover, we performed a summary descriptive analysis of the tetanus cases by reporting month to identify any seasonal pattern. The analysis was conducted in Microsoft Excel for 365 and RStudio version 2024.12.1. *p*-values less than 0.05 were considered statistically significant.

### 2.2. Tetanus Surveillance in Romania

In Romania, both tetanus and neonatal tetanus are case-based, mandatory notifiable diseases. A passive surveillance system is in place. All healthcare providers have the legal responsibility to report a case meeting the case definition for tetanus or neonatal tetanus. The case definition used in Romania is a nationally adapted version of the EU case definition from 2018 [24]. Although laboratory diagnosis of tetanus presents limitations, the Romanian case definition incorporates specific laboratory criteria to facilitate the evaluation of antibody titres in suspected cases, thereby contributing to evidence-based vaccination policy development [25].

For children and adults, a suspected case of tetanus is defined by the presence of at least two of the following three clinical manifestations: painful muscle contractions, especially of the masseter and neck muscles, resulting in characteristic facial spasms known as trismus and *rissus sardonicus*; painful muscle contractions of the trunk or limbs in the absence of another identified cause; and, respectively, generalized spasms or frequent episodes of opisthotonus.

For newborns, a suspected case of tetanus is defined by the development of an inability to suck between days 3 and 28 associated with muscle contractions, seizures, or both, without another obvious cause, in an infant who had a normal suck reflex and cries in the first 2 days of life.

The laboratory criteria include at least one of the following elements: a serum titre of anti-tetanus antibodies considered non-protective (determination performed prior to the administration of specific therapy) in a clinically suspected case of tetanus or seroconversion indicating a significant increase in the titre of specific antibodies over time.

Epidemiological criteria are not applicable.

Concerning classification in children and adults, cases are classified into probable and confirmed cases. The possible case category is not applicable.

In newborns, a probable case is defined either by meeting the clinical criteria or by the attending physician’s suspicion of possible neonatal tetanus. A probable case is considered confirmed if the newborn dies between days 3 and 28 of life or if one of the laboratory criteria is met. Serological investigation of the patient must be performed before the administration of specific therapy and is mandatory, as is the evaluation of the mother’s specific immunological status in cases of neonatal tetanus.

For every suspected case of tetanus, healthcare providers must fill out the unique reporting sheet, which follows the informational flow described in the legislation; additionally, an enhanced surveillance form is completed by the Public Health Authorities in collaboration with the treating physician [6,26,27].

### 2.3. Ethical Approval

In this study, we present aggregated epidemiological data obtained as part of national surveillance efforts for which ethical approval is not required under the Romanian legal framework.

## 3. Results

### 3.1. Number of Tetanus Cases and Annual Incidence Rate

From 2010 to 2023, 97 tetanus cases were reported in Romania, 85 (87%) being classified as confirmed, while 12 cases (13%) were classified as probable. The median number of cases per year was 6.5, with an interquartile range (IQR) of 2 (5–7). The maximum number of cases (20) was reported in 2011, and the minimum (3) in two of the analysed years (2014, 2020) (Figure 1). The number of tetanus cases decreased annually on average by 7% (IRR: 0.93; 95% CI: 0.88–0.98; *p* = 0.005).

The average tetanus incidence during the study period was 0.03 per 100,000 inhabitants (95% CI: 0.02–0.04; range: 0.01–0.09) (Figure 2). However, the decline in incidence of 7% is not supported by the statistical analysis (IRR: 0.93; 95% CI: 0.44–1.98; *p* = 0.9).

### 3.2. Seasonal Pattern of Tetanus in Romania

During the study period, tetanus cases were reported each month, with variations from year to year. The highest number of cases was reported in July (17; 18%) and the lowest number during February (2; 2%) and March (2; 2%) (Figure 3).

### 3.3. Demographic Characteristics of Tetanus Cases

Rural residents accounted for most of the tetanus cases (74 cases, 76%), with approximately three times the number of tetanus cases recorded in urban areas. In 2012, 2016, and 2021, all reported cases involved rural patients. In contrast, the number of urban cases exceeded that of rural cases in 2010 (5 cases vs. 1 case) and 2020 (2 cases vs. 1 case).

The average tetanus incidence among rural residents was 0.05 per 100,000 inhabitants (95% CI: 0.03–0.08; range: 0.01–0.17) compared with 0.01 per 100,000 inhabitants among urban residents (95% CI: 0.007–0.019; range: 0.00–0.04). The linear regression model revealed that rural residents had a 4% higher incidence of tetanus than urban residents (*p* < 0.001). Notably, in 2011, the tetanus incidence among rural residents was approximately 7-fold higher than that among urban residents.

Most cases were reported in males (70; 72%). The male-to-female sex ratio was 3:1. Tetanus cases were consistently reported among males during the entire analysed period, with five years showing exclusively male cases. However, the number of female cases surpassed that of male cases in two of the studied years (2015 and 2022). The average tetanus incidence among males was 0.05 per 100,000 inhabitants (95% CI: 0.03–0.06; range: 0.03–0.12) compared with 0.02 per 100,000 inhabitants among females (95% CI: 0.008–0.026; range: 0.00–0.05). The linear regression model showed males having a 3% higher tetanus incidence than females (*p* = 0.003). In 2012, the tetanus incidence among males was approximately 6-fold higher than in females.

The highest number of cases was reported in people aged 65 and over (*n* = 39, 40%), with an average incidence rate of 0.08 per 100,000 inhabitants (95% CI: 0.04–0.12; range: 0.00–0.24). However, the highest average incidence rate was recorded among those aged 1 to 14 years old (0.09 per 100,000 inhabitants, 95% CI: 0.06–0.13; range: 0.00–0.20), with 26 cases recorded in this age group (27%). In contrast, the lowest average incidence rate was recorded in the age group 15–24 years old (0.01 per 100,000 inhabitants; 95% CI: 0.00–0.047; range: 0.00–0.10), with 4 cases recorded in this age group (4%). The linear regression model showed an 8% lower incidence of tetanus cases among those aged 15 to 24 years old and among those aged 25 to 64 years old (*p* < 0.001) compared with those aged 1 to 14 years old. The incidence among those aged 65 and over was 1% lower than that among those aged 1 to 14 years old, but the difference was not statistically significant (*p* = 0.5).

Additionally, the annual incidence of tetanus cases slightly decreased in almost all age groups, but it was not statistically significant: 15 to 24 years old—15% decrease, IRR: 0.846, 95% CI: 0.19–3.58, *p* = 0.8; 25 to 64 years old—10% decrease, IRR: 0.903, 95% CI: 0.302–2.7, *p* = 0.9; 65 years and older: 13% decrease, IRR: 0.874, 95% CI: 0.53–1.44. In contrast, in the age group 1 to 14 years old, the incidence of tetanus increased by 4% (IRR: 1.036; 95% CI: 0.68–1.59), but this finding was not statistically significant (*p* = 0.8) (Table 1).

No cases of neonatal tetanus or cases under the age of 1 were identified during the study period.

The average tetanus incidence by county was 0.03 per 100,000 inhabitants (95% CI: 0.02–0.04; range: 0.00–0.12), with the highest incidence recorded in Dolj (0.12 per 100,000 inhabitants; 95% CI: 0.003–0.239; range: 0.00–0.83); in this county, 12 tetanus cases were reported during the studied period. Nine counties reported no tetanus cases (Brăila, Brașov, Covasna, Hunedoara, Maramureș, Sălaj, Sibiu, Vâlcea, and Vrancea) (Figure 4).

Of the 90 cases for which data on severity was available, 67 (74.4%) had a severe form of disease. Severe clinical manifestations were recorded every year. In 2013, 2015, 2016, and 2019, all reported cases presented with severe forms of the disease.

Between 2010 and 2023, a total of 47 deaths caused by tetanus were recorded in Romania out of 97 reported cases. The peak year for tetanus cases also recorded the highest number of fatalities. In 2013, all tetanus cases died (six cases). In comparison, no deaths were reported in 2014, 2020, or 2021 (Figure 1).

The number of tetanus deaths decreased annually on average by 14% (IRR: 0.86; 95% CI: 0.79–0.93; *p* < 0.001). The average case fatality ratio was 42% (95% CI: 25.62–57.92; range: 0–100), showing a decreasing trend of 9% by year (IRR: 0.91; 95% CI: 0.89–0.93) (Table 1).

The average crude mortality rate was 0.015 per 100,000 inhabitants (95% CI: 0.008–0.022; range: 0.00–0.049), showing the same decreasing trend of 14% by year, which was not supported by the statistical analysis (IRR: 0.086; 95% CI: 0.27–2.79; *p* = 0.8). Similar numbers of deaths were recorded among males and females (25 and 22), with an average mortality rate of 0.016 (95% CI: 0.009–0.024; range: 0.00–0.045) among males and 0.014 (95% CI: 0.004–0.023; range: 0.00–0.06) among females (*p* = 0.4), but the average case fatality ratio was higher among females than males (47.1% vs. 33.8%). The distribution of deaths by age group showed the highest number of deaths among those aged 65 years and over (38; CFR: 74%; mortality rate: 0.06 per 100,000 inhabitants), followed by those aged 25–64 years (28; CFR: 39%; mortality rate: 0.01 per 100,000 inhabitants). In the age group 1–14 years old, 6 deaths were recorded (CFR: 23%; mortality rate: 0.02 per 100,000 inhabitants), while among those aged 15 to 24 years, 1 death was recorded (CFR: 25%; mortality rate: 0.00 per 100,000 inhabitants). The linear regression showed no statistically significant differences in the mortality rates among age groups.

The tetanus mortality rate showed variations depending on the county (Figure 5). The highest value (represented in red) was recorded in Dolj (0.08 per 100,000 inhabitants), with a case fatality ratio of 67%. A moderate mortality rate (represented in orange) was recorded in five counties, three of which are neighbouring counties in the east of the country. Most counties presented a low mortality rate (represented in yellow). On the other hand, there were seven counties that had reported cases of tetanus, but no deaths in the 14 years of study (represented in green). In the nine counties marked in white, no cases of tetanus were reported during the analysed period.

### 3.4. Vaccination Status of the Tetanus Cases

Vaccination status was available for 85 cases: 61 cases (72%) were unvaccinated, 12 (14%) had received an incomplete schedule, while for the remaining 12 cases (14%), the vaccination history could not be determined.

### 3.5. The Impact of the Vaccination Coverage on Tetanus Incidence

In Romania, the average vaccination coverage at 12 months with 3 doses of tetanus-containing vaccines over the study period was 68% (95% CI: 60.8–76.0; range: 27–83), 71% in urban areas (95% CI: 63.7–77.6; range: 32.6–85.6), and 65% (95% CI: 56.9–73.9; range: 19.5–80.6) in rural areas. There were no statistically significant differences in vaccination coverage trends over time. The linear regression models showed that a 1% increase in vaccination coverage leads to a decrease in tetanus incidence by 0.001 units (*p* = 0.003). The decrease in tetanus incidence in rural areas was of 0.002 units (*p* < 0.001), while in urban areas it was of 0.0002 units (*p* = 0.4).

## 4. Discussion

During the study period, Romania’s reported tetanus case count was substantial, consistently positioning it within the top five European countries. In 2011, with 20 cases, Romania recorded the second highest number of cases in Europe, surpassed only by Italy [16]. The trend of tetanus cases in Romania is in accordance with the European situation, with the highest number of cases reported in 2011 (149 cases), followed by a progressive decline. The lowest number of tetanus cases was recorded in 2020 and 2021, which was probably influenced by the COVID-19 pandemic [16].

Based on this study’s findings, the tetanus trend in Romania seems to be stable over time, despite a reduction in the number of cases and deaths during the studied period.

In developing this study’s aim and methodology, we started from the hypothesis according to which, in recent years, the epidemiological and demographic characteristics of tetanus cases were influenced by the existing vaccination policies and coverages. Therefore, we expected to have tetanus cases in all the age groups, but the highest incidence was expected in the older age groups due to waning immunity, and the lowest incidence among children and adolescents, who should be protected by vaccination. However, these hypotheses were only partially confirmed. As we expected, among seniors, the tetanus incidence was high, explained by a reduction in the protective antibody titres over time and the absence of booster doses during adulthood. Our findings are supported by a study conducted in thirteen Italian regions between June 2019 and May 2020 [28].

However, unexpectedly, the highest incidence rate was recorded among the youngest age group, among those who should have benefited from vaccination. According to the data available on the ECDC website for the period 2010–2022, the proportion of cases in the youngest age group was more than 8 times greater in Romania (27%) than in Europe (3.4%) [29]. This high percentage is recorded in a context where, in Romania, vaccination coverage in children aged 12 months according to the vaccination schedule was below the 95% target throughout the entire analysed period, consistently below the European level, and exhibiting significant fluctuations, in contrast to the stable European trend [21,23,27].

No cases of neonatal tetanus were identified in Romania or in any other EU member state during the entire study period, as confirmed by the ECDC and national data. In addition, no tetanus cases were reported among PWIDs in Romania; however, at the European level, this remains a recognized risk group [30].

The distribution of tetanus cases by residence shows a significantly higher incidence in rural areas (76%). This may suggest a considerable impact of exposure, a low level of awareness regarding tetanus risks, and limited access to preventive and curative medical services compared with the urban population [31].

These findings highlight the need to promote more the recently adopted policies of fully reimbursed diphtheria–tetanus–pertussis booster doses every 10 years for adults and the need to implement actions aimed at increasing the vaccination uptake among children, especially in rural areas.

The recommendation is supported by the linear regression results, showing that a 1% increase in vaccination uptake leads to a decrease of 0.001 units of tetanus incidence, and particularly in rural areas, where the highest incidence of tetanus cases was recorded during the study period.

Data analysis by county showed Dolj county having the highest incidence, a county known as an agricultural area with a high risk of occupational exposure of the resident population. Two other counties with increased incidence rates are Vaslui and Neamț, which are among the counties with the lowest socio-economic status in the country [32].

However, during the 14 years studied, nine counties did not report any cases. A possible explanation is that, except for Brăila County, all the other eight counties are in the Carpathian Mountain range, and agricultural activities are less common.

In Romania, tetanus incidence was higher in males than in females; these findings are in accordance with the global tetanus epidemiology, explained by occupational exposure and the higher vaccination rates among females due to the global initiatives aiming to eliminate neonatal tetanus [33].

Regarding the seasonal pattern, although cases were reported each month, most cases were recorded in July, which is an expected result, considering the risk of exposure. On the other hand, a less anticipated result was the large number of cases in December. One explanation could be a possible delay in establishing the diagnosis and reporting the case, with the exposure being at the end of autumn, during October–November, when intensive agricultural activities are carried out.

The severity of the cases reported, as well as the high case fatality rate (three times higher than the European average of 14.4%) underscore the need for developing actions aimed at raising awareness regarding tetanus risk factors and protective measures (including vaccination and proper management of tetanus-prone wounds) among the general population, but also among healthcare workers, especially those providing primary healthcare.

As expected, tetanus fatality is higher among seniors. These results are in accordance with the results reported in other studies. For example, a global study on tetanus reported higher incidence and mortality rates in people aged 35 years and over, while a study coming from Turkey reported a higher case fatality ratio among those aged 60 and older, but it was not statistically significant, considering the sample size (*n* = 53 patients) [9,33]. Moreover, according to the US surveillance data collected from 2013 to 2022, all tetanus-related deaths occurred among patients aged over 60 years [34]. The associated comorbidities could be also a possible explanation for the severe evolution of the disease in older people. A study from the Philippines demonstrated hypertension as increasing the hazard of death by 4.5 times (*p* = 0.004), possibly linked to the interaction between an already compromised cardiovascular status and autonomic dysfunction, usually associated with severe tetanus [35]. Our study results showed a higher case–fatality ratio in females than in males, which contrasts with the existing data typically showing either equal rates or higher fatality in males [36,37]; however, these results should be interpreted cautiously due to the small sample size.

The current study highlighted that tetanus remains a relatively rare infection in Romania, but it follows a constant trend, emphasizing the importance of:○Public awareness campaigns addressing tetanus risk factors and protective measures;○Awareness campaigns for healthcare professionals on tetanus epidemiology, surveillance, and control;○Continuous monitoring of the availability of tetanus toxoid-containing vaccines and antitoxin.

### 4.1. Study Limitations

The limitations of this study were mainly derived from data completeness and availability, considering the extended study period, as well as variance in surveillance methodologies at the national and European levels [38]. Additionally, information on the type of wound, incubation period, and tetanus management were not considered. According to the ECDC report, underreporting of mild tetanus cases in partially immunized patients, or the possibility that severe cases are more likely to be identified and recorded, cannot be excluded [16].

### 4.2. Future Perspectives

The evolution of tetanus in Romania should be attentively monitored in the future, especially for assessing the impact of vaccination policies for adults on tetanus incidence in older age groups. Another potential future direction could be an evaluation of the knowledge and practices according to the existing tetanus management guidelines, with particular emphasis on the most-affected regions.

## 5. Conclusions

In conclusion, the current study revealed the fact that tetanus continues to represent a challenge for Romania despite the availability of tetanus toxoid-containing vaccines, highlighting not only the need to increase the uptake among those eligible, but also the need to ensure proper and prompt management of tetanus-prone wounds.

The surveillance of tetanus is an important tool for evaluating the regional and demographic trends in order to tailor adequate interventions.

## Figures and Tables

**Figure 1 microorganisms-13-01654-f001:**
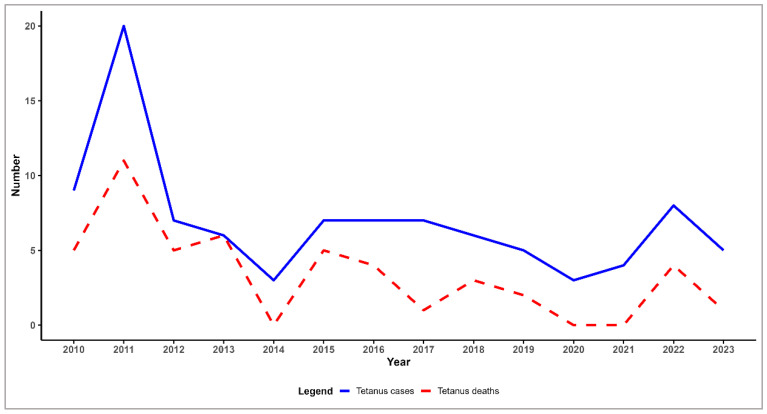
Distribution of tetanus cases and deaths by year in Romania, 2010–2023.

**Figure 2 microorganisms-13-01654-f002:**
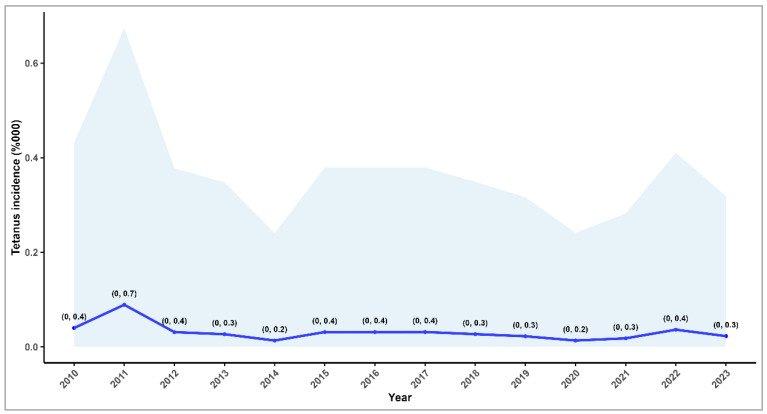
Tetanus incidence by year, with 95% confidence intervals, Romania, 2010–2023.

**Figure 3 microorganisms-13-01654-f003:**
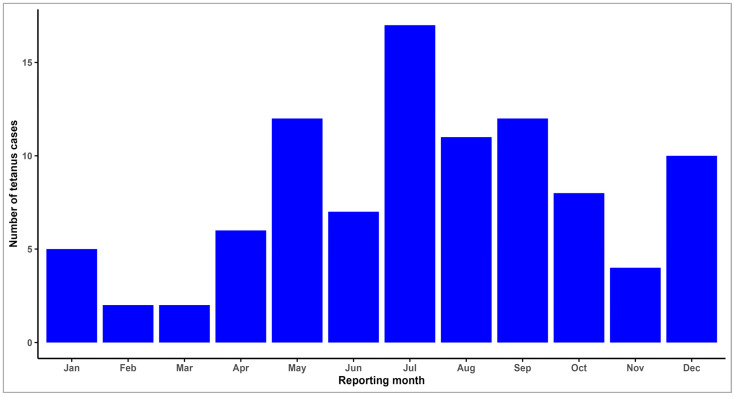
Distribution of tetanus cases by reporting month, Romania, 2010–2023.

**Figure 4 microorganisms-13-01654-f004:**
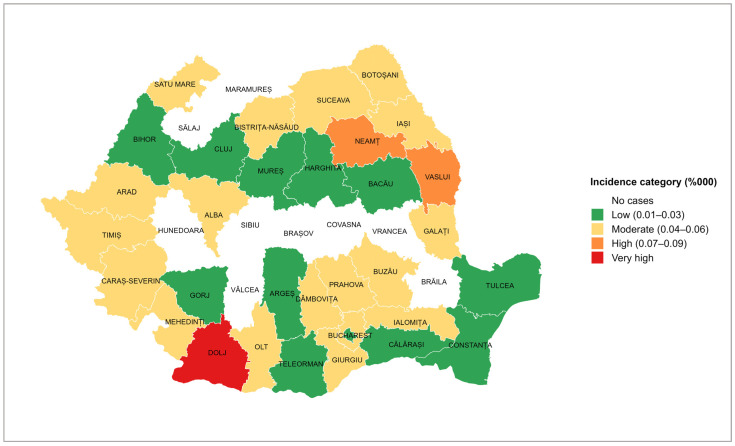
Distribution of tetanus incidence rate by county, Romania, 2010–2023.

**Figure 5 microorganisms-13-01654-f005:**
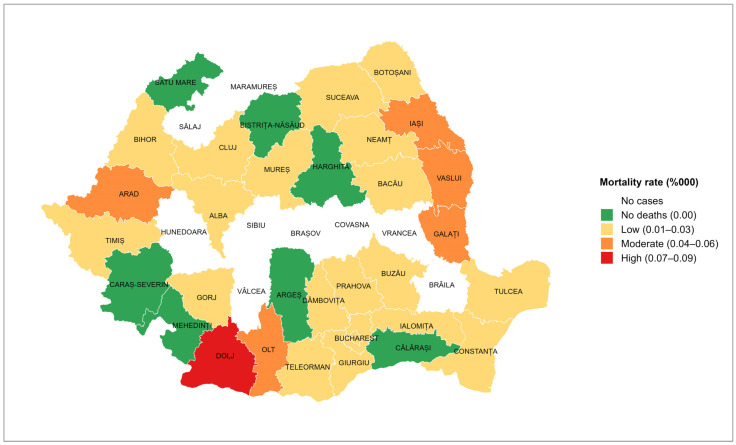
Distribution of the tetanus mortality rate by county, Romania, 2010–2023.

**Table 1 microorganisms-13-01654-t001:** Main epidemiological indicators of tetanus morbidity and mortality, Romania, 2010–2023.

Year	Incidence Rate by Residency (per 100,000 Inhabitants)		Incidence Rate by Sex (per 100,000 Inhabitants)		Incidence Rate by Age Group (per 100,000 Inhabitants)				Case Fatality Rate **(%)**
	Urban	Rural	Female	Male	1–14 Years	15–24 Years	25–64 Years	65 Years and over	
**2010**	0.04	0.04	0.00	0.08	0.05	0.00	0.02	0.15	55.56
**2011**	0.02	0.17	0.05	0.13	0.10	0.10	0.05	0.24	55.00
**2012**	0.00	0.07	0.01	0.05	0.00	0.00	0.02	0.12	71.43
**2013**	0.01	0.05	0.02	0.04	0.00	0.00	0.01	0.15	100.
**2014**	0.01	0.02	0.00	0.03	0.10	0.00	0.01	0.00	0.
**2015**	0.02	0.04	0.04	0.03	0.20	0.00	0.00	0.09	71.43
**2016**	0.00	0.07	0.03	0.04	0.10	0.00	0.02	0.06	57.14
**2017**	0.02	0.05	0.00	0.06	0.20	0.00	0.00	0.08	14.29
**2018**	0.01	0.05	0.03	0.03	0.05	0.00	0.03	0.03	50.00
**2019**	0.01	0.04	0.01	0.04	0.15	0.00	0.01	0.03	40.00
**2020**	0.02	0.01	0.00	0.03	0.15	0.00	0.00	0.00	0.
**2021**	0.00	0.04	0.00	0.04	0.05	0.04	0.02	0.00	0.00
**2022**	0.02	0.05	0.04	0.03	0.10	0.00	0.02	0.08	50.00
**2023**	0.01	0.04	0.02	0.03	0.05	0.00	0.00	0.10	20.00

## Data Availability

The original contributions presented in this study are included in the article. Further inquiries can be directed to the corresponding author.

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
