# Peer review of "Tetanus in Romania—Trends and Challenges"

_microorganisms, 2025, doi:10.3390/microorganisms13071654_

Round 1
Reviewer 1 Report
Comments and Suggestions for Authors
This paper analyzes medical record data of relatively high (for Europe) tetanus incidence and fatalities in Romania from 2010 to 2023. The paper is mostly straightforward and well-written. The data are somewhat anecdotal because of the small sample size. The data are medically relevant and should encourage higher vaccination rates and better vaccine information distribution in Romania and other European countries where incidence of tetanus is somewhat high (i.e., Italy). The paper hopefully can counteract the anti-vaccination propaganda movements in some countries. Fatalities from tetanus are around 50% of detected cases.
Some minor points:
“The highest mortality rate was recorded in Dolj (0.08%000), with a case fatality ratio of 67% (Figure 5).” Figure 5 is not sufficiently described in the text.
“Figure 5. Distribution of the tetanus mortality rate by county, Romania, 2020-203.” Not sure what was intended here. Maybe 2020-2023? Maybe 2010-2023?
IRR is not defined. “IRR, or internal rate of return, is a metric used in financial analysis to estimate the profitability of potential investments.” From Wikipedia. Does not appear to fit the paper. Not sure what IRR means in statistical analysis.
Author Response
Review report 1
Dear Reviewer,
Thank you for your insightful feedback! Below is the list of changes made to the manuscript as advised:
- “The highest mortality rate was recorded in Dolj (0.08%000), with a case fatality ratio of 67% (Figure 5).” Figure 5 is not sufficiently described in the text.
Thank you! We improved the description as it follows:
“Tetanus mortality rate showed variations depending on the county (Figure. 5) The highest value (represented in red) was recorded in Dolj (0.08 per 100,000 inhabitants %000), with a case fatality ratio of 67%. A moderate mortality rate (represented in orange) was recorded in five counties, three of which are neighbouring counties in the east of the country. Most counties presented a low mortality rate (represented in yellow). On the other hand, there were seven counties that had reported cases of tetanus, but no deaths in the 14 years of study (represented in green). In the nine counties marked in white, no cases of tetanus were reported during the analyzed period.”
2. “Figure 5. Distribution of the tetanus mortality rate by county, Romania, 2020-203.” Not sure what was intended here. Maybe 2020-2023? Maybe 2010-2023?
Thank you! There was a typing error, which was corrected. The period is 2010-2023
3. IRR is not defined. “IRR, or internal rate of return, is a metric used in financial analysis to estimate the profitability of potential investments.” From Wikipedia. Does not appear to fit the paper. Not sure what IRR means in statistical analysis.
Thank you! We added the explanation of IRR abbreviation in the Materials and methods section: “Incidence rate ratio (IRR) was calculated to highlight trends in tetanus situation over time”
Reviewer 2 Report
Comments and Suggestions for Authors
See comments to the manuscript

Author Response
Review report 2
Dear Reviewer,
Thank you for your insightful feedback! Below is the list of changes made to the manuscript as advised:
1.Mistakes
Page 8 Table 90
Figure 5 2020-223
Thank you! There were typing errors, which were corrected.
2.It is unclear, what does it mean? Please, explain:
Segment 3.3.
Line 5 - 0.05%000
line 6 - 0.01%000
line15 - 0.05%000
line 16 - 0.02%000
line 20 - 0.08%000
line 22 - 0.09%000
line 24 - 0.01%000
Page 7
Line 5 from the bottom -0.03%000
There are the same 0.0%000 meanings in subheadings of the Table 1. These methods of presenting results should be explained. How it was happened.
Thank you for your comments! Tetanus is still a rare disease in Romania, with a low number of cases recorded annually and low incidence rates, which explain the figures. Your feedback was very well appreciated and we have worked to improve clarity of the manuscript by replacing %000 with per 100,000 inhabitants. However, we will keep the indicators or rates for better reflecting the true burden of tetanus and for allowing comparisons.
- Should be explained as well:
In Table 1:
Line 2013 showes 100 % of case fatality, but line 2014 showes 0% of case fatality. How it was happened? Everythings is absolutely unclear. But the numbers in both lines look like the same
Thank you! In Table I, we aimed to summarize the key epidemiological indicators describing the burden of tetanus, respective incidence rates (by urbanity and sex) and the case fatality rate. The case fatality rate was calculated as the number of deaths divided by the number of cases in that year, multiplied by 100. In both years, cases occurred among males and females, and in both rural and urban areas; therefore, we calculated incidence rates for these groups. However, in 2014 no deaths were reported, resulting in a case fatality rate of 0%, whereas in 2013 all reported cases died, leading to a case fatality rate of 100%. We did not have any additional clinical information available which could explain the severity.
The information on tetanus deaths is available in the manuscript:
“In 2013, all tetanus cases died (six cases). In comparison, no deaths were reported in 2014, 2020 and 2021 (Figure 1).”
- In Discussion: the following phrases contradict each other:
Page 10
In Romania, the tetanus incidence was higher in males than in females, these findings are in accordance with the global tetanus epidemiology, explained by occupational exposure and lower vaccination rates, but in contrast with the data reported at the European level [16] [33].
Thank you for your comment! We rephrased the paragraph as it follows:
“In Romania, tetanus incidence was higher in males than in females, these findings are in accordance with the global tetanus epidemiology, explained by the occupational exposure and higher vaccination rates among females due to the global initiatives aiming to eliminate neonatal tetanus [33].”
Page 11
severe tetanus [35]. Unexpectedly, the case-fatality ratio was higher in females than in males, which contrasts with the existing data typically shown either equal rates or higher fatality in males [36] [37].
Thank you for your comment! We rephrased the paragraph as it follows:
“Our study results showed a higher case-fatality ratio in females than in males, which contrasts with the existing data typically shown either equal rates or higher fatality in males [36] [37], however these results should be cautiously interpreted due to the small sample size.”
- The following cases of mild tetanus cases are of interest. Could it be explained more in detail? In 4.1 segment:
Furthermore, we cannot exclude underreporting of mild tetanus cases in partially immunized patients, or the possibility that severe cases are more likely to be identified and recorded [16].
What are partially immunized patients? How are they created and do they pose a danger to humans as carriers of a weak but still effective infection?
Thank you for your comment! We rephrased the paragraph from the study limitations as it follows:
“According to the ECDC report, underreporting of mild tetanus cases in partially immunized patients, or the possibility that severe cases be more likely to be identified and recorded cannot be exclude [16].”
Tetanus transmission was explained in the introduction:
„ Although it is a vaccine-preventable disease described since ancient times, without human-to-human transmission, seemingly paradoxically, tetanus continues to remain a public health problem”
All the modified sections can be traced directly in the manuscript.
Also, I would like to confirm that all authors have reviewed and approved the revised version of the manuscript.